# Exploring the value in variations of the Relative Income Price (RIP) for calculating cigarette affordability: An illustration using Malaysia

Ridzuan Kunji Koya[1], J. Robert Branston[1,2]*, Allen W. A. Gallagher[1]

1 Tobacco Control Research Group, Department for Health, University of Bath, Bath, United Kingdom,
2 School of Management, University of Bath, Bath, United Kingdom

* J.R.Branston@bath.ac.uk

**Data Availability Statement:** All data used herein is publicly available and can be obtained from the sources listed within the manuscript or the attached data supplementary file.

## Abstract

The relationships between cigarette affordability, consumer income levels and distribution, and tax increases are complex and underexplored. This study investigates different ways of calculating the Relative Income Price (RIP) measure of affordability using Malaysia as a case study. We calculate cigarette affordability in Malaysia between 2009–2019 using government data, and multiple RIP variants. The conventional RIP calculation relies on 2,000 sticks and GDP (henceforth standard RIP). We explore that and other variants that use annual cigarette consumption estimates and/or proportions of various financial measures of wealth in both rural and urban areas. Our findings indicate broadly consistent trends in cigarette affordability across all methods. From 2009 to 2012, there was a slight decrease in the percentage of wealth required to purchase cigarettes, followed by an increase in 2015 and 2016, and then another decline, suggesting a recent trend toward increased affordability. Using the standard RIP method, 0.9 percentage points(pp) more of per capita GDP was required between 2009 and 2016, but, by 2019 it was 0.1pp less than in 2016. However, Household Income Per Capita (HIPC) and Household Expenditure Per Capita (HEPC) provide a more nuanced perspective on cigarette affordability compared to GDP per capita, as they reveal larger shifts in affordability. The conventional 2,000 sticks method using HIPC from 2009 to 2016 indicated 0.3pp more of income was required to purchase cigarettes, but by 2019, it was 1.0pp less than in 2016. Using HIPC with actual consumption estimates, smokers required approximately 0.9pp more of average income to purchase cigarettes between 2014 and 2016, but 2.5pp less from 2016 to 2019. Actual consumption estimates offer insight into smokers' ability to offset higher purchase costs by adjusting consumption patterns without quitting. We conclude that to address issues related to cigarette affordability, the Malaysian government should consider increasing tobacco tax vis-à-vis income growth.

**Funding:** JRB and AWAG receive funding from Bloomberg Philanthropies, as part of the Bloomberg Initiative to Reduce Tobacco Use. RK is supported by a University of Bath research Studentship and by a Central Bank of Malaysia PhD Scholarship. The funders had no role in study design, data collection and analysis, decision to publish, or preparation of the manuscript.

**Competing interests:** JRB owns 10 shares in Imperial Brands for research purposes. The shares were a gift from a public health campaigner and are not held for financial gain or benefit. All dividends received are donated to health related charities, and proceeds from any future share sale or takeover will be similarly donated. This does not alter our adherence to PLOS ONE policies on sharing data and materials.

## Introduction

The affordability of tobacco products is a crucial factor in determining tobacco use. This affordability is influenced by both the prices of tobacco products and consumers' income levels [1]. One widely acknowledged method for reducing the affordability of tobacco, and consequently curbing its consumption, is the implementation of higher tobacco taxes to raise the price of tobacco products [2–5]. For instance, a 2017 study conducted in Malaysia estimated that a 5.9% tax-induced increase in tobacco prices would lead to a 3.4% decrease in consumption [6]. Furthermore, a study in 2021, focusing on the relationship between tobacco taxes, prices, and consumption in both High-Income Countries (HIC) and Low- and Middle-Income Countries (LMIC) concluded that price increases resulting from tax changes have been highly effective in reducing cigarette consumption on a global scale [1]. This study highlighted the need for targeted policies and further research, especially in LMIC, where economic growth has influenced cigarette consumption dynamics differently from HIC. One of the issues highlighted was that rapid income growth can offset the impact of tax-induced price increases of tobacco products, potentially making products like cigarettes affordable for consumers even when taxes are raised [7, 8].

In order to explore the crucial role of cigarette affordability and the effectiveness of higher cigarette taxes in reducing consumption, various measures of cigarette affordability have been suggested. Such measures allow the exploration of affordability and its changes, both over time and between countries [9–11]. They include: Minutes of Labour (MoL), expressed as working minutes needed to purchase a pack of local brand cigarettes [5]; the cigarette price/daily income ratio (CPDIR) to determine the percentage of daily income needed to buy a pack of cigarette [12]; and the most widely used, the Relative Income Price (RIP), which is the proportion of a nation's annual Gross Domestic Product (GDP) per capita needed to purchase 2,000 cigarette sticks (adopted as a round number approximation of typical per capita consumption levels) [11, 13].

The RIP method, originally developed by Blecher and van Walbeek (2004) [13], offers several advantages relative to the other approaches. Firstly, GDP is considered to be a good economic indicator of overall living standards within a country [14, 15]. Secondly, GDP data is commonly available, and hence the RIP is a convenient measure to calculate. Lastly, every country calculates per capita GDP using a broadly consistent methodology, which allows global comparison of cigarette affordability between countries and over time [11]. However, despite the RIP being widely used and having several advantages, the conventional approach of calculating it using GDP as a measure of wealth fails to always accurately reflect the income of those people who smoke, thereby presenting a major drawback [14, 16]. For instance, the method has been criticised for its inability to accommodate differences in income distributions [14, 17]. Furthermore, people who smoke often respond to price increases by changing their smoking habits, rather than quitting (e.g., using less tobacco in each hand rolled cigarette), which is not something that can be accounted for in the conventional RIP calculation, given the assumption of 2,000 sticks [8, 10, 14, 16–18].

Given these shortcomings, previous studies have suggested alternative versions of the RIP, such as moving away from GDP as a measure of consumer income in order to better explore real-world affordability [1, 8, 9, 14, 16]. These concerns are relevant in both high income (HIC) (e.g., United States, United Kingdom, Australia, and Canada) [19] and low- and middle-income countries(LMIC), like Indonesia [17]. For instance, in China, household income per capita (HIPC) has been utilised by the World Bank as a more fitting indicator of consumer income, rather than GDP [14]. The reason being that the government and state-owned companies hold a significant portion of the country's wealth which is incorporated in GDP but does

not necessarily translate into wealth that benefits the general population, including people who smoke [14, 16]. Similarly, in Russia due to the substantial inequalities in income distribution, HIPC across distinct income groups has been used (in place of a single GDP per capita measure) as a more reliable metric for assessing affordability among different socio-economic strata [8]. The suggestion is that using HIPC allows an examination of cigarette affordability for individuals in different income groups, hence analysing the impact of cigarette tax increases on the cost of cigarettes consumed by each income group, which cannot be calculated using GDP [8]. In the case of Indonesia, the use of household expenditure per capita (HEPC) has been suggested due to high income inequality and a large informal labour market among its population [17]. HEPC is proposed as it represents the money the consumers have available to spend on tobacco products, capturing actual disposable income in families as the significant informal labour makes HIPC less reliable in Indonesia, due to the high proportion of unreported income [17].

It would therefore seem that moving away from the standard RIP to calculate cigarette affordability may be appropriate in certain circumstances. However, despite these suggestions, it is unclear what impact using alternative measures of wealth (e.g., GDP, HIPC, and HEPC) has, as there has been no comparative evaluation incorporating all these measures in the same setting. As such, the question of what measure is most suitable remains unanswered. To fill this knowledge gap, we calculate multiple measures of the RIP to determine cigarette affordability in Malaysia and evaluate the differing insights gained. To achieve this, we also employ a unique approach that incorporates actual cigarette consumption patterns, thereby taking into consideration the distinctions that exist between rural and urban areas. This distinction between areas is valuable as the data shows that rural areas are relatively low-income groups while urban areas are the relatively high-income groups. This method not only enhances our understanding of cigarette affordability but also allows us to explore the impact of consumer behaviour from different incomes groups in mitigating adverse changes in affordability.

Our chosen case study is of particular interest due to the possibility of LMICs, including Malaysia, experiencing accelerated economic growth compared to high-income nations [20]. This economic growth can potentially diminish the effectiveness of tobacco taxes in reducing cigarette affordability over time. For instance, in the decade leading up to September 2021, Malaysia consistently achieved positive overall GDP growth, averaging between 4% and 6% annually [21]. However, amidst this overall economic prosperity, data published by the Malaysian government identifies significant disparities in household income levels and smoking behaviours between rural and urban areas [20, 22, 23]. These disparities are of great relevance as they highlight the intricate challenges associated with implementing effective tobacco control measures through taxation where both social and economic factors are at play. Moreover, similarly to Indonesia, research indicates that Malaysia struggles with a considerable amount of unreported consumer income [24, 25]. This unreported income substantially influences household expenditure, which, in turn, will directly impact smoking behaviours and consumption patterns. Malaysia therefore a fitting setting to conduct a comprehensive and in-depth analysis of economic growth, household income/expenditure, and smoking patterns to fully understand the dynamics of cigarette affordability. This understanding is crucial for developing effective strategies to reduce smoking rates and enhance public health in Malaysia and elsewhere.

Malaysia has introduced a series of tobacco excise tax revisions since the enactment of a cigarette minimum price law (MPL) in 2010, which implemented a minimum cigarette selling price of MYR10.00 ($\approx$ USD2.20) and banned the sale of cigarettes in less than 20 stick packs [26, 27]. However, since 2010, tobacco tax increases in Malaysia have been infrequent as there are no routine adjustments to account for the impact of inflation [28, 29]. As of the time of

writing, the government has yet to revise the tobacco tax rate since an approximate 25.9% increase in retail cigarette prices in 2015, when the system changed from being a mixed excise tax system (specific and *ad valorem* taxes) to an entirely specific one. Therefore, Malaysia's infrequent tobacco excise tax revisions make it an especially pertinent case for examining the issues surrounding cigarette affordability and the calculation of the RIP.

The contribution of this study is to explore: i) the impact of using different financial measures of wealth to calculate cigarette affordability and changes in affordability levels within the period of study; ii) the affordability as experienced by the people who smoke in Malaysia using conventional fixed consumption, and for the first time, using actual cigarette consumption values, including splits between rural and urban areas; and iii) the effectiveness of government's tax policy on cigarette affordability in recent years, and to make suggestions for improvement. Such findings provide detailed data for Malaysia, and useful insights for how similar work could be carried out elsewhere, hence providing wider lessons to other countries.

## Methodology

### Affordability and Relative Income Price (RIP) measures

To assess the affordability of cigarettes in Malaysia, we employed the Relative Income Price (RIP) measure using different financial measures of wealth. We initially calculated the conventional RIP using the widely established approach of considering the cost of 2,000 sticks as a proportion of Gross Domestic Product (GDP) per capita. We then calculated the RIP using Household Income Per Capita (HIPC) and Household Expenditure Per Capita (HEPC) in place of GDP. This allowed us to compare the RIP using different measures of wealth and hence to see how the RIP is affected by the choice of financial measure.

Recognising the limitation of the conventional RIP calculation which presumes a fixed annual cigarette consumption of 2,000 sticks and may not represent the true habits of people who smoke in Malaysia, we then integrated yearly cigarette consumption estimates (measured in sticks) into our calculations. We explored these measures of consumption in both rural and urban areas. Furthermore, we explored actual consumption values in two ways. We firstly explored the habits of people who smoke (i.e. mean consumption of reported smokers only) and then secondly, tobacco consumption within the overall population (i.e. mean consumption covering those who smoke and those who do not). The former allowed us to gain a more accurate understanding of cigarette affordability for people who smoke in Malaysia, while the latter allows a more direct comparison with the traditional RIP (as that considers the population overall too). By adopting this version of the RIP we are essentially looking to explore the affordability of the habit of smoking rather than the affordability of cigarettes which is what the RIP with the fixed 2,000 stick consumption level is doing. By comparing the results between the different RIP series this holistic approach provides a comprehensive perspective on the overall economic impact of cigarettes, thereby facilitating an assessment of affordability and potential socioeconomic implications for the general population.

### Data sources

In order to calculate the RIP using various financial measures of wealth (GDP, HIPC, and HEPC), we used actual yearly cigarette price data obtained from the Malaysian Ministry of Domestic Trade and Consumer Affairs, covering 2009–2019 [30]. Information on the yearly population size, yearly GDP, aggregate household income (available for the years 2009, 2012, 2014, 2016, and 2019), and aggregate household expenditure (available for the years 2014, 2016 and 2019) were all sourced from the official Malaysian government website [23].

**Table 1. Yearly GDP, HIPC and HEPC per capita (All in current MYR).**

| Yearly in MYR | Gross Domestic Product (GDP) Per Capita | Household Income Per Capita (HIPC) Overall | Household Income Per Capita (HIPC) Urban | Household Income Per Capita (HIPC) Rural | Household Expenditure Per Capita (HEPC) Overall | Household Expenditure Per Capita (HEPC) Urban | Household Expenditure Per Capita (HEPC) Rural |
|---|---|---|---|---|---|---|---|
| 2009 | 25,385.74 | 10,733.28 | 12,546.72 | 6,786.72 | | | |
| 2010 | 28,731.33 | 12,681.75 | 14,650.86 | 7,881.09 | | | |
| 2012 | 29,760.51 | | | | | | |
| 2013 | 31,611.17 | | | | | | |
| 2014 | 32,967.53 | 17,137.68 | 19,068.84 | 10,691.16 | 9,985.08 | 10,942.33 | 6,784.19 |
| 2015 | 37,738.09 | | | | | | |
| 2016 | 38,860.09 | 20,364.84 | 22,451.76 | 12,758.04 | 11,803.92 | 12,883.90 | 7,975.61 |
| 2017 | 40,620.80 | | | | | | |
| 2018 | 42,115.99 | | | | | | |
| 2019 | 41,321.53 | 24,310.80 | 26,569.20 | 15,396.96 | 13,950.72 | 15,126.15 | 9,347.69 |

*Sources*: [22, 23, 30]

The yearly cigarette consumption estimates, including differences between rural and urban areas, were developed from recently published data by Koya et al. in 2022, calculated using the Malaysian government's official National Health and Morbidity Survey (NHMS) which is conducted every four years (further details provided below) [22, 29]. The appropriateness of using these estimates (as opposed to other ones published at about the same time [31]) has been discussed elsewhere and these have been found to be the best available [28].

Our study focuses exclusively on Factory Made cigarettes (FM) (including *kretek;* Indonesian clove cigarettes) as these account for 99% of tobacco products consumed in Malaysia (based on the excise duty paid from 2011 to 2019 [32]).

### GDP per capita calculation

Table 1 provides information on the yearly GDP per capita. Since GDP per capita is not available from the government's website, we manually calculated GDP per capita by dividing yearly GDP values with the yearly population size (see the overall detailed information and calculation on the yearly GDP per capita in S1 Table).

### Household Income Per Capita (HIPC) and Household Expenses Per Capita (HEPC)

To obtain the HIPC and HEPC (see Table 1), we divided the aggregate gross household income (for HIPC) and aggregate household expenditure (for HEPC) by the mean size of a household for each of the studied years (as shown in S2 Table). This was done for rural and urban areas individually, and also the overall combined value for the country as a whole. For example, the average gross household income of the urban population for 2009 (MYR 4,705) is divided by the mean size of household in 2009 (4.5) to obtain the HIPC of MYR 1,045.56 (see S2 Table). It was necessary to use household gross income (income before tax and any benefits) for the HIPC calculation since the data on household disposable income deductions (that allow for tax and benefits) was not published for years before 2016. For the HEPC, our calculations only cover 2014 onwards since household expenditure data was not available prior to that. The gap between HIPC and HEPC consists of unreported income, income tax, social security scheme payments, *zakat* (state-mandated Islamic almsgiving), and savings. The use of

the urban and rural split to explore different consumer incomes/wealth levels was utilised as such information by decile or other such breakdown of levels was not available for the years being considered.

## Cigarette consumption estimates

We developed cigarette consumption estimates following the methodology of Koya et al. (2022) [29] which were derived from the NHMS survey, covering 2011 to 2019. As the daily smoking frequency in the survey was captured as a percentage of people who smoke in different consumption ranges (e.g., 1 to 4 cigarettes or 5 to 9 cigarettes per people who smoke per day), consumption point estimates were generated using a weighted average approach. One notable difference herein is the estimates were developed for rural and urban areas separately, as well as overall estimates. To offer sensitivity analysis for our calculations, as per Koya et al. we produced four different consumption estimates (ultra-low, lower, central, and upper) for both areas, in line with the previous methodology for doing so [29] (see cigarette consumption estimates in the rural and urban areas in S3 and S4 Tables).

## Endogeneity between price and consumption levels

When exploring the relationship between cigarette consumption and price simultaneously, potential endogeneity issues may arise (given that consumption levels are in part—but not solely—determined by price). Endogeneity is a common concern, particularly in econometric analyses, because it can lead to biased numerical estimations. In such cases, the coefficients derived in an analysis may be inaccurate and not reflect true relationships within the data [33, 34]. However, since our study does not involve regression analysis or similar numerical techniques, this concern does not apply.

Versions of the RIP featuring consumption estimates will explore the combined impact of consumer behaviour, prices, and incomes/wealth, whereas the traditional RIP versions using 2,000 sticks will explore the impact of prices and incomes only. As such the differences between the two different types of RIP will explore the extent to which consumers are able to mitigate or take advantage of the changes to price and income taking place. The two types of RIP can therefore be considered to be the affordability of cigarettes (2,000 stick RIP) and the affordability of the habit of smoking (Consumption estimate RIP). As noted above, such insights are potentially valuable in a market like Malaysia, particular given the market has a large illicit tobacco sector, which has notably increased in market share during the period of analysis (2011–40.7%; 2015–55.0%; 2019–69.9%) [28].

By offering the RIP series using consumption volume estimates, we look to offer a range of insights into the issue of tobacco affordability. This most notably means how consumer behaviour adapts to price changes over time and the use of the urban and rural areas within this also allows this to vary across different demographics of people who smoke. Such an approach provides greater understanding beyond a single elasticity figure, which is how consumer responses to price is traditionally explored. In this regard it is worth noting that the most recent elasticity estimates for Malaysia were published in 2013 [35], and were based on an analysis conducted in 2009, and the country/market has experienced considerable changes since then, especially in terms of income growth and the rate of illicit tobacco products.

## RIP calculations

In short, we explore affordability using two types of RIP: (i) the conventional RIP of 2,000 sticks per year (henceforth 2000RIP) as a proportion of the different financial measures of wealth (per capita GDP, HIPC, and HEPC) from 2009 to 2019; and (ii) a newly developed

version using actual cigarette consumption estimates (ultra-low, lower, central, and upper) (ConsumptionRIP) as a proportion of the different financial measures of wealth (per capita GDP, HIPC and HEPC) from 2011 to 2019. The data for cigarette consumption estimates only became accessible in 2011, preventing a direct comparison of ConsumptionRIP with the 2000RIP that commenced in 2009.

To measure the 2000RIP, we first determined the price of 2,000 cigarette sticks for each of the study years (the price of 100 packs of 20 sticks). In the context of affordability, selecting the most economical option serves as an indicator of the product's accessibility to consumers [11]. Hence for the purpose of this calculation we specifically concentrated on price data from the lowest legal cigarette price segment (sub-premium quality cigarettes) in both rural and urban areas. We then calculated the price of 2,000 sticks as a percentage of the per capita GDP, HIPC, and HEPC. For the ConsumptionRIP, we derived estimates by multiplying consumption figures by the stick price and subsequently calculated the values as a percentage of per capita GDP, HIPC, and HEPC. This calculation was conducted separately for the two approaches to dealing with actual consumption: individuals who smoke (total consumption divided by the smokers' population) and the overall population (total yearly consumption divided by the total population). The results of the RIP for the latter are presented in S5 & S6 Tables.

## RIP nomenclature

In naming the different versions of the RIP, we adopt the following convention as outline in Table 2 below.

## Results

### Affordability trends using GDP per capita

We present the results in percentages for both types of RIP (2000RIP and ConsumptionRIP), indicating the proportion of these financial measures required to purchase cigarettes, with higher figures indicating less affordability (see Tables 3 & 4). Although we explored multiple series of consumption estimates to derive the RIP, this results section focuses solely on the

**Table 2. Nomenclature of the different versions of the RIP.**

| Type of Cigarette Consumption Measure Used | Type of Financial Measure of Wealth Used | Type of Income Distribution Used | Label Used |
|---|---|---|---|
| Conventional (2,000 sticks) | GDP | Overall | 2000RIP-GDP-Overall |
| | HIPC | Overall | 2000RIP-HIPC-Overall |
| | | Rural | 2000RIP-HIPC-Rural |
| | | Urban | 2000RIP-HIPC-Urban |
| | HEPC | Overall | 2000RIP-HEPC-Overall |
| | | Rural | 2000RIP-HEPC-Rural |
| | | Urban | 2000RIP-HEPC-Urban |
| Actual Consumption Estimates* | GDP | Overall | ConsumptionRIP-GDP-Overall |
| | HIPC | Overall | ConsumptionRIP-HIPC-Overall |
| | | Rural | ConsumptionRIP-HIPC-Rural |
| | | Urban | ConsumptionRIP-HIPC-Urban |
| | HEPC | Overall | ConsumptionRIP-HEPC-Overall |
| | | Rural | ConsumptionRIP-HEPC-Rural |
| | | Urban | ConsumptionRIP-HEPC-Urban |

Note: *These estimates can be done for four versions of different consumption levels. Unless stated otherwise, the results herein are based on the central consumption estimates

**Table 3. Affordability calculation using conventional method (2000RIP).**

| | | | 2009 | 2010 | 2011 | 2012 | 2013 | 2014 | 2015 | 2016 | 2017 | 2018 | 2019 |
|---|---|---|---|---|---|---|---|---|---|---|---|---|---|
| Gross Domestic Product (GDP) Per Capita | Overall | Central Estimates | 3.1% | 3.0% | 2.9% | 2.9% | 3.3% | 3.6% | 4.1% | 4.0% | 3.8% | 3.8% | 3.9% |
| Household Income Per Capita (HIPC) | Overall | | 7.3% | 6.7% | | | | 7.0% | | 7.6% | | | 6.6% |
| | Urban | | 6.2% | 5.8% | | | | 6.3% | | 6.9% | | | 6.0% |
| | Rural | | 11.5% | 10.8% | | | | 11.2% | | 12.1% | | | 10.4% |
| Household Expenses Per Capita (HEPC) | Overall | | | | | | | 12.0% | | 13.1% | | | 11.5% |
| | Urban | | | | | | | 11.0% | | 12.0% | | | 10.6% |
| | Rural | | | | | | | 17.7% | | 19.4% | | | 17.1% |

Source: Author's own calculation

central estimates. For a comprehensive view of the other consumption scenarios, please consult S7 and S8 Tables.

The affordability of cigarettes in Malaysia exhibited a generally consistent pattern across all versions of the RIP, with each measure varying and fluctuating around distinct points on the affordability scale. Nevertheless, our principal focus in this analysis has been on examining the trend in these changes and, consequently, shifts in affordability levels rather than focusing solely on the absolute values or percentage points of the RIP themselves.

Overall, there was a minor decrease in the percentage of wealth needed to buy cigarettes (indicating increased affordability) from 2009 to 2012, followed by an increase with a peak in 2015 and 2016 (indicating decreased affordability), and then another decline, suggesting a more affordable trend in smoking in the more recent study years (2016 to 2019). For instance, according to the conventional 2000RIP-GDP-Overall benchmark, 3.1% of the GDP was needed to buy cigarettes in 2009. This RIP reached its highest point in 2015 (4.1%) before gradually decreasing to 3.9% in 2019, indicating a trend of increased affordability for cigarettes after 2015 (see Table 3). Likewise, the ConsumptionRIP-GDP-Overall also follows a similar pattern, where the percentage of wealth needed to purchase cigarettes peaked in 2015 (10.0%), before declining in the subsequent years (from 10.0% to 8.7%), again suggesting an increasing affordability in cigarettes in the recent years (2016 to 2019) (see Table 4).

## Affordability trends using different financial measures of income

The pattern of cigarette affordability as measured by HIPC and HEPC also broadly reflects the trends seen in the conventional GDP measure. For instance, in 2014, 7.0% of income was

**Table 4. Affordability calculation using actual consumption estimates (ConsumptionRIP).**

| | | Central Estimates | 2009 | 2010 | 2011 | 2012 | 2013 | 2014 | 2015 | 2016 | 2017 | 2018 | 2019 |
|---|---|---|---|---|---|---|---|---|---|---|---|---|---|---|
| Gross Domestic Product (GDP) Per Capita | Overall | | | | 7.5% | 7.5% | 8.4% | 9.0% | 10.0% | 9.5% | 8.9% | 8.7% | 8.7% |
| Household Income Per Capita (HIPC) | Overall | | | | | | | 17.3% | | 18.2% | | | 14.8% |
| | Urban | | | | | | | 15.6% | | 16.4% | | | 13.4% |
| | Rural | | | | | | | 27.8% | | 29.6% | | | 24.6% |
| Household Expenses Per Capita (HEPC) | Overall | | | | | | | 29.8% | | 31.4% | | | 25.9% |
| | Urban | | | | | | | 27.2% | | 28.6% | | | 23.5% |
| | Rural | | | | | | | 43.8% | | 47.3% | | | 40.5% |

Source: Author's own calculation

Note: Results in both Tables 3 and 4 are presented as percentages where the higher figures denote a higher proportion of the financial measures of wealth (GDP, HIPC and HEPC) required to purchase cigarettes. The higher the percentage, the less affordable tobacco to be.

required to purchase cigarettes based on the 2000RIP-HIPC-Overall (Table 3). This percentage increased to 7.6% in 2016 but later decreased to 6.6% in 2019, also indicating a slight enhancement in affordability towards the end of the study years. Similarly, the percentage of expenditure required to buy cigarettes was 12.0% in 2014 according to the 2000RIP-HEPC-Overall (Table 3). This figure rose to 13.1% in 2016 but then declined to 11.5% in 2019, again suggesting an overall increase in affordability towards the conclusion of the study period.

However, looking at the detail, there are some differences in the percentage of wealth required (and changes therein) when comparing the different 2000RIP calculations. For example, the purchase of cigarettes required 3.1% of GDP in 2009 based on the 2000RIP-GDP-Overall, which increased to 3.9% in 2019. Conversely, the purchase of cigarettes required 7.3% of income in 2009 based on 2000RIP-HIPC-Overall, which actually decreased to 6.6% in 2019. A similar pattern of a decline in financial wealth required is also observed in HEPC and from 2014 to 2019, which is again the opposite result to the GDP calculation.

## Disparities in rural and urban areas and changes in affordability levels

Our findings highlighted that employing both 2000RIP and ConsumptionRIP in conjunction with different financial measures of wealth has produced broadly similar results in assessing the overall affordability trend. This trend remains uniform for both rural and urban areas, as illustrated in Figs 1 and 2. However, notable disparities do again exist in some of the fine detail, particularly concerning changes in affordability during the years 2014 to 2016 and 2016 to 2019 in these rural and urban regions.

When we employed 2000RIP with HIPC or HEPC as alternatives to GDP for measuring affordability changes in both rural and urban areas, the outcomes were noticeably different. Indeed, these alternate measures suggest that the amount of financial wealth required is notably larger than is suggested by standard versions of the RIP using GDP. Furthermore, the shift in affordability became more pronounced when we integrated the ConsumptionRIP with HIPC and HEPC.

For instance, when analysing cigarette affordability using 2000RIP-HIPC-Urban, the changes in affordability percentages appear relatively moderate (see Table 5). In urban areas, the percentage of income required to purchase cigarettes increased from 6.3% in 2014 to 6.9% in 2016, reflecting a difference of 0.6 percentage points (a 9.52% change in RIP, and hence

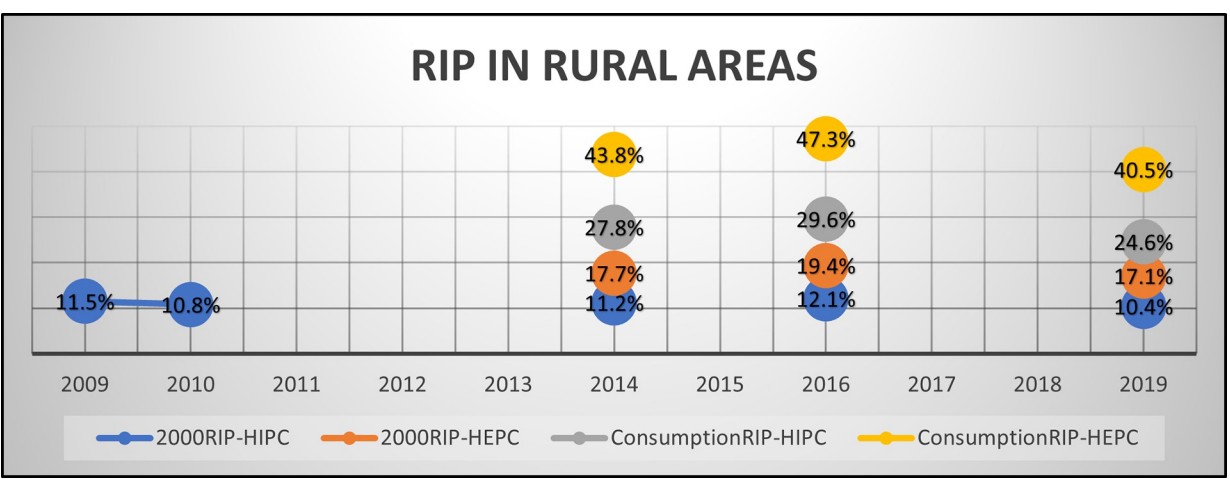

**Fig 1. RIP in rural areas.**

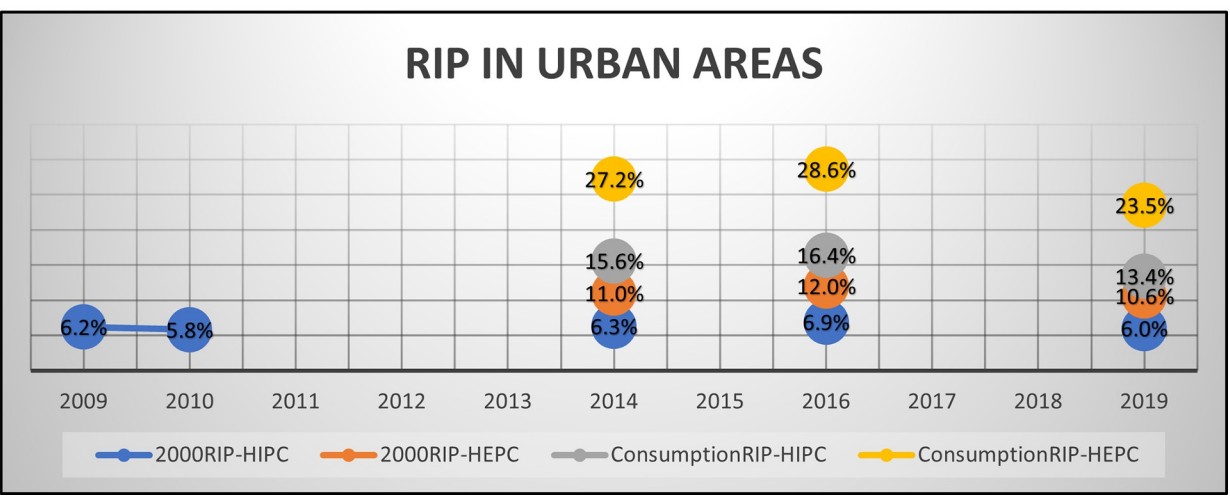

**Fig 2. RIP in urban areas.**

affordability). Subsequently, it decreased to 6.0% in 2019, representing a difference of 0.9 percentage points (and hence a -13.04% change in affordability). Similarly, in rural areas, our 2000RIP-HEPC-Rural analysis indicated a discernible increase in the percentage of expenditure needed for cigarettes, rising from 17.7% in 2014 to 19.4% in 2016—a difference of 1.7 percentage points and a 9.61% change in affordability. However, it then decreased to 17.1% in 2019 –a difference of 2.3 percentage points and -11.86% affordability change.

However, when utilising actual cigarette consumption estimates in both rural and urban areas, more notable changes in cigarette affordability were observed. According to ConsumptionRIP-HIPC-Urban, the percentage of income required for cigarettes increased from 15.6% in 2014 to 16.4% in 2016, reflecting a difference of 0.8 percentage points with 5.13% change in affordability. However, it decreased to 13.4% in 2019, representing a notable difference of 3.0 percentage points, an -18.29% affordability change (see Table 5). In rural areas, the percentage of expenditure required for cigarettes using ConsumptionRIP-HEPC-Rural experienced a

**Table 5. Change In affordability with various measures of income and consumption estimates.**

| Measures of affordability | Income (HIPC)/ expenditure (HEPC) required to purchase cigarettes (%) | | 2014–2016 percentage point difference (B–A) = (C) (%) | Change in affordability (C / A) = (D) (%) | Income (HIPC)/ expenditure (HEPC) required to purchase cigarettes | 2016–2019 percentage point difference (E–B) = (F) (%) | Change in affordability (F / B) = (G) (%) |
|---|---|---|---|---|---|---|---|
| | 2014 (A) | 2016 (B) | | | 2019 (E) | | |
| 2000RIP-HIPC-Urban | 6.3% | 6.9% | 0.6% | 9.52% | 6.0% | -0.9% | -13.04% |
| 2000RIP-HIPC-Rural | 11.2% | 12.1% | 0.9% | 8.04% | 10.4% | -1.7% | -14.05% |
| 2000RIP-HEPC-Urban | 11.0% | 12.0% | 1.0% | 9.09% | 10.6% | -1.4% | -11.67% |
| 2000RIP-HEPC-Rural | 17.7% | 19.4% | 1.7% | 9.61% | 17.1% | -2.3% | -11.86% |
| ConsumptionRIP-HIPC-Urban | 15.6% | 16.4% | 0.8% | 5.13% | 13.4% | -3.0% | -18.29% |
| ConsumptionRIP-HIPC-Rural | 27.8% | 29.6% | 1.8% | 6.47% | 24.6% | -5.0% | -16.90% |
| ConsumptionRIP-HEPC-Urban | 27.2% | 28.6% | 1.4% | 5.15% | 23.5% | -5.1% | -17.83% |
| ConsumptionRIP-HEPC-Rural | 43.8% | 47.3% | 3.5% | 7.99% | 40.5% | -6.8% | -14.38% |

substantial rise from 43.8% in 2014 to 47.3% in 2016, indicating a difference of 3.5 percentage points and 7.99% change in affordability. However, it then decreased to 40.5% in 2019, representing a substantial difference of 6.8 percentage points with -14.38% affordability change.

When looking the affordability as calculated using mean tobacco use for the entire country population (see S6 Table), we find a relatively lower level of the RIP. This suggests that cigarettes are more affordable when considering the overall population, in contrast to calculations based solely on the smokers' population. For instance, our ConsumptionRIP-HEPC-Rural, when applied to the overall population, revealed that the percentage of expenditure required to purchase cigarettes was 6.4% in 2014, compared to 43.8% within the smokers' population. By 2016, these figures had increased to 6.9% and 47.3% respectively. Such differences explain why the results herein might look considerably different from previously published studies on first inspection (see for example, the results in a paper by Tan et al. (2020), which measured the budget share of tobacco products in Malaysia using the Household Expenditure Surveys (HES) and found that the overall budget share for tobacco products was 4.6%) [36]. Nevertheless, when considering the changes in the two affordability measures between dates, the aforementioned illustration shows there is broad consistently between the measures. Between 2014 and 2016 there is a change in affordability of 7.8% (6.4% to 6.9%) using the overall population but 7.99% within the smokers' population (43.8% to 47.3%).

### Other notable finding

Our findings into the relationship between GDP per capita and cigarette affordability, specifically between 2014 and 2015, has also revealed an interesting trend. Notably, during these two years, there was a substantial growth in GDP per capita as compared to other years, rising from MYR 32,967.53 in 2014 to MYR 37,738.09 in 2015 (14.49% increase). However, contrary to the expected outcome of increased affordability with economic growth, our findings indicate a different pattern. The conventional RIP in 2014 resulted in a recorded value of 3.6% of GDP, which surprisingly increased to 4.1% in 2015, indicating decreased affordability. Equally unexpected, when utilising actual consumption estimates, the RIP was valued at 9.0% in 2014, and this value increased to 10.0% in 2015, also indicating a substantial decrease in affordability. This implies the growth in income was more than offset by the increases in tobacco taxation.

## Discussion

This paper has explored cigarette affordability in Malaysia from 2009 to 2019 using various versions of the RIP method, including adopting multiple financial measures of wealth (GDP, HIPC, and HEPC), conventional yearly consumption of 2,000 cigarette sticks a year, and estimates of actual cigarette consumption in both rural and urban areas.

In this section we provide a broad overview of the different insights gained from our findings on the affordability of cigarettes in Malaysia. We separate the discussion into three parts: first, the wider methodological implications, itself divided into three subsections–(i) methodological implications, (ii) consumption estimates for monitoring smoking behaviour, and (iii) cigarette prices, socioeconomic factors and an endogeneity issue; second, the implications for Malaysia specifically; and finally, the implications for wider relevant issues, including illicit cigarette purchases, non-price factors, and concerns around public health. We provide associated policy recommendations at the end of each section.

### Methodological implications–General Interpretation of results

In measuring cigarette affordability, the utilisation of GDP per capita is a straightforward and convenient approach. It allows for a relatively simple means of making general comparisons

and identifying overarching trends, both within a specific country and between different nations. However, our results show that while GDP per capita provides a useful starting point, it comes with inherent limitations when evaluating the affordability of cigarettes in countries like Malaysia.

GDP fails to offer a nuanced breakdown of income distribution among various segments of the population, including urban and rural areas in the case herein. This limitation is substantial, as it means that GDP may not comprehensively reflect the disparities in wealth and income levels within a country. Our findings have clearly shown that using alternative measures such as HIPC and HEPC can offer different insights into cigarette affordability, as both provide a more granular representation of the income/expenses of a particular household compared to using GDP to calculate RIP. These alternative measures matter because they offer a more nuanced understanding, capturing variations in household income and expenditure that GDP alone does not account for- a feature that can be beneficial not only in our context but also in offering insight on similar disparities in income and expenses in various countries [23, 37, 38].

For instance, our study showed that the affordability figures changed differently when we allow HIPC and HEPC to replace the GDP. To illustrate, when using our 2000RIP-GDP-Overall benchmark, it becomes apparent that 4.0% of GDP was needed to purchase cigarettes in 2016, which decreased slightly to 3.9% in 2019, marking a mere 0.1 percentage point difference. Conversely, when we calculated affordability using 2000RIP-HIPC-Overall, 7.6% of income was required to buy cigarettes in 2016, which reduced to 6.6% in 2019, indicating a more substantial 1.0 percentage point difference compared to the GDP-based calculation. Furthermore, when considering 2000RIP-HEPC-Overall, 13.1% of expenditure was required to buy cigarettes in 2014, and this decreased to 11.5% in 2019, representing a notable 1.6 percentage point difference compared to the GDP-based assessment. Therefore, these divergent changes in affordability figures underscore how HIPC and HEPC provide different and unique perspectives on the perception of cigarette affordability changes [10, 16], which cannot be gleaned from GDP data alone.

Hence, our results suggest that the benefits of exploring subtle changes in affordability using alternative measures need to be weighed against the challenges involved in calculating them. In light of resource constraints and the need for a practical starting point, we suggest that policymakers begin with the standard RIP calculation using GDP. This provides a solid foundation for assessing cigarette affordability. As resources allow, policymakers can then expand their approach to encompass multiple measures and indicators, perhaps periodically examining them to gain a more comprehensive understanding of the impact of policies and economic conditions on cigarette affordability. This phased approach ensures that, even with limited resources, policymakers can make informed decisions regarding tobacco control measures and their economic implications.

## Consumption estimates for monitoring changes in smoking behaviour

Our findings have clearly illustrated that employing consumption estimates offers unique insights compared to the conventional benchmark of 2,000 sticks. For instance, when using our 2000RIP-HIPC-Overall measure, 7.6% of income is needed to buy cigarettes in 2016, a figure that decreased to 6.6% in 2019, marking a notable -1.0 percentage point difference. In contrast, our ConsumptionRIP-HIPC-Overall measure presented a different perspective. It demonstrated that individuals who smoke required a more substantial 18.2% of their income to purchase cigarettes in 2016, which decreased to 14.8% in 2019, indicating a -3.4 percentage point difference, which is notably larger than the figure based on the 2,000-stick benchmark.

As such, we recommend simultaneous monitoring of actual consumption estimates alongside the standard 2,000 sticks approach. This dual approach ensures that vital information is not overlooked in the assessment of cigarette affordability, particularly when smoking habits are experiencing subtle or ongoing changes, including reduction in daily cigarette intake or switching to more cost-effective alternatives like roll-your-own (RYO) cigarettes [18].

Our research has also revealed that when incorporating actual consumption data in both rural and urban areas, the change in cigarette affordability becomes more apparent. Specifically, it is much more noticeable that affordability decreases when considering the actual cigarette consumption estimates from 2016 to 2019. For example, in rural areas, the expenditure required to purchase cigarettes as measured using HEPC (2000RIP-HEPC-Rural) showed a less notable decrease (from 19.4% to 17.1%, approximately -2.3 percentage points) when compared to using actual consumption estimates (ConsumptionRIP-HEPC-Rural) (from 47.3% to 40.5%, approximately -6.8 percentage points). A similar trend was observed in urban areas as well.

These examples clearly demonstrate the impact of incorporating actual consumption estimates on understanding the real-world dynamics of cigarette affordability. They allow the fluctuations in affordability as experienced by smokers to be made more transparent, thereby allowing policymakers to gain better insights into the variations in cigarette affordability over time. This comprehensive approach facilitates the formulation of targeted and evidence-based interventions to tackle smoking prevalence effectively and promote public health across diverse socioeconomic groups in Malaysia.

## Cigarette consumption, prices, and socioeconomic factors

In our study, we aimed to attain a more comprehensive understanding of the genuine impact of cigarette affordability by delving into the intricate interplay between price and consumption, including in urban and rural areas. Our findings hold significance, as they suggest that in the presence of price increases, some individuals who smoke likely counteract declines in affordability by altering their smoking behaviour.

For example, the 2000RIP-HEPC-Urban in our study shows a relatively static affordability trend, as cigarettes respectively required 11.0%, 12.0%, and 10.6% of expenditure in the years 2014, 2016, and 2019. However, there was a subtle change in the details of the trend and how the affordability level changed when the consumption level is allowed to adjust, given the ConsumptionRIP-HEPC-Urban shows the people who smoke respectively required 27.2%, 28.6%, and 23.5% of their expenditure to purchase cigarettes. Since there was an absence in cigarette tax and price increases, the substantial 5 percentage point decrease observed between 2016 and 2019 in this RIP might be partially attributed to lower consumption volumes, as evidenced by the drop in mean consumption from 4,751 to 4,434 sticks (as well as increasing income levels). Thus, our findings indicate that the actual consumption estimates may reflect the true cost of smoking and the changes in smoking behaviour over time, which the standard consumption of 2,000 sticks a year does not offer.

Moreover, our findings also emphasise the substantial influence of taxation, as considerable income growth measured by GDP per capita was counteracted by considerable increase in tobacco taxes, impacting the affordability. Despite robust GDP per capita growth from MYR 32,967.53 in 2014 to MYR 37,738.09 in 2015, the anticipated increase in cigarette affordability did not occur. Notably, the percentage of GDP required to purchase cigarettes increased from 3.6% to 4.1%, contradicting the assumption that higher GDP should decrease the percentage of income needed for cigarettes, indicating increased affordability. This unexpected outcome coincided with an overall reduction in annual cigarette consumption from the year 2014 to

2015, declining from 4,955 to 4,866 sticks per person. Furthermore, a 2015 tax increase that resulted in an approximate 25.9% increase in retail cigarette prices likely offset the potential rise in cigarette affordability, underscoring the pivotal role of tobacco taxation in shaping the policy landscape surrounding the cigarette affordability.

In the course of our analysis, we acknowledged a possible minor endogeneity issue when using consumption estimates within the RIP given the inherent link between cigarette consumption and pricing, but which interestingly, adds a unique dimension to our study. For instance, the Malaysian government, in its regulatory capacity, sets cigarette prices across all tobacco products through its minimum price rule and decisions over taxes. In response, consumers adapt their consumption behaviours and the true impact of price changes on consumer behaviour remains uncertain until the consumers alter their smoking habits. It is worth noting that this consumer response may not occur immediately, indicating a sequential cause-and-effect relationship. However, it should also be noted that consumers might change their tobacco consumption patterns in response to factors other than price and income levels (e.g. perceptions of health risks). As such our twin exploration of both actual consumption estimates and the conventional 2000 sticks helps us generate a more comprehensive understanding of changes in the affordability of cigarettes, and hence ultimately, public health outcomes.

## Implications for Malaysian policy

Our findings indicate that by the end of the study period (with the exception of most of the RIP variants that that use GDP per capita), Malaysia's cigarette affordability was nearly the same as or lower than it was at the starting year of our study (e.g. 2000RIP-HIPC-Overall in 2009 (7.3%) and 2019 (6.6%)). Hence, we observed that tobacco tax has not increased sufficiently to offset both income growth and the shifts in tobacco usage patterns (see Table 5) [1, 11, 14, 39]. Therefore, the Malaysian government should consider raising the excise tax, most especially in the absence of cigarette excise tax revisions for the past eight years. During the period between 2010 and 2015, our results have clearly shown that Malaysia's enactment of the minimum price law [26] and intermittent increases in excise tax duty played a significant role in decreasing cigarette affordability. However, since 2016, in almost all version of the RIP we have explored, we have observed a decrease in the RIP, indicating an increase in affordability. This observation underscores the importance of taxation in influencing affordability, especially in those contexts where the resulting price increase can effectively counterbalance any rise in income and/or behavioural changes [40].

Malaysia has also yet to introduce a tobacco tax policy that considers tobacco taxation over time, which has likely hindered the Malaysian government's efforts to reduce cigarette affordability and consumption effectively and consistently over the past decade. Such an approach may be advantageous when one considers the actions and situations of the UK, New Zealand, and Australia as being world-leaders in tobacco control [41–43], and also Thailand in the context of an LMIC, as a country with successful tobacco control policies.

In the UK, for instance, there is political commitment to a tax escalator which increases cigarette excise tax by at least 2% above inflation each year. Furthermore, such consistent increases were found to have successfully helped to lower tobacco affordability by 27% in 2016 compared to 2006 [41]. In light of such practices and other tobacco control measures, the prevalence of smoking in the UK has reduced from 30% of the population in the year 2000 to 15% in 2020 [41]. In New Zealand, the government, until recently, employed a tax escalator of 10% above inflation as part of their tobacco control strategies, and have now moved towards other measures to achieve their tobacco end game in 2025 [42, 44]. In Australia, the practice is to uplift excise in line with average wage inflation, while there were also further increases up to

12.5% annually between May 2016 and 2020, which collectively lowered affordability and helped to sustain a decline in smoking rates [45–47]. In Thailand, the high level of tobacco tax at 75% of the retail price and consistent tobacco tax increases over the years have helped to reduce the smoking prevalence from 31.7% in 2000 to 22.1% in 2020 [48, 49]. A significant smoking reduction has also been noticed particularly among low-income individuals, who are often the most price-sensitive and likely to be deterred by higher prices [50, 51].

As such, the Malaysian government should consider adopting a multi-year tobacco tax policy that effectively reduces cigarette affordability and consequently, consumption. One possible strategy is to establish a tax escalator based on inflation and/or changes in average wages. Australia's wage adjustment approach may be the most appropriate for Malaysia, given the country's income levels that tend to increase at a faster rate than inflation [46]. Furthermore, exploring how tobacco tax increases might be combined with other mutually supporting proposals (e.g., cessation support programmes) could lead to even more desirable outcomes [46].

## Illicit cigarette purchases and non-price factors

An increase in tobacco taxes is often blamed for fueling the illicit market, particularly by the Tobacco Industry (TI), although studies have shown that it is not the case in several countries including Malaysia [26, 38, 52, 53].

In recent local news reports, the ease of obtaining illicit products in rural areas has been highlighted, where they are often openly sold in rural groceries due to a lack of adequate enforcement controls [54]. This wide-spread availability, combined with the possibility of those on lower incomes obtaining illicit cigarettes at a lower cost than their legal equivalents, does create an incentive for people who smoke to switch to illicit product [55, 56]. However, the motivation to switch to illicit cigarettes due to price differences has not materially changed over time. Illicit cigarettes consistently remain cheaper than legally sold cigarettes by a similar margin, mainly attributed to the illicit market's tendency to closely track legal prices [57, 58]. Additionally, switching to illicit cigarettes is further influenced by a range of other non-price factors, such as the ease of obtaining illicit supplies, which is itself linked to the presence of high levels of corruption, and the effectiveness of border enforcement [59]. These additional factors can significantly shape consumers' choices and behaviours related to cigarette purchases, making it important for policymakers to consider a comprehensive approach in addressing the issue of illicit cigarette trade most especially when taxes are increased.

Implementing effective tobacco tax policies in Malaysia requires a comprehensive approach that addresses both legislative and enforcement challenges. Tobacco tax increases in tMalaysia do not entail an adjustment for inflation. Tobacco taxation adjustments require a new law each time they are changed, meaning that each change, even one just to account for inflation, requires a clear policy choice by the government. This legislative requirement underscores the importance of deliberate policy-making to ensure that tax changes are both effective and sustainable.

As such, to efficiently implement tobacco tax increases, it is imperative for the country to implement/enhance measures that will support those tax changes, such as, effective enforcement measures which target the supply chain of illicit tobacco products, including production, distribution, and sales [60–62]. Such measures may include strengthening customs controls, increasing penalties for those involved in the illicit trade, and improving international co-operation among law enforcement agencies [63, 64]. Supply-side approaches are complementary to demand-side approaches such as targeted by tax increases [57].

In line with the above, Malaysia should consider ratifying the World Health Organisation (WHO) FCTC Protocol to Eliminate Illicit Trade in Tobacco Products ('the Protocol') [65].

One of the key components of the Protocol is the implementation of tracking and tracing systems for tobacco products, which would require Malaysia to establish a robust tracking and tracing mechanism that ensures the transparency and accountability of the tobacco supply chain. This system would help enable the government and relevant authorities to track the movement of legal product, identify any diversion into the illicit market, and identify the origin of illicit tobacco products. It would also help in detecting counterfeit or smuggled tobacco products, making it easier for law enforcement agencies to take targeted actions against those involved in illicit trade [65].

## Strengths and limitations

This study contributes (to the best of our knowledge) the first estimates of cigarette affordability using a diverse range of RIP measures. It evaluates the significance of incorporating different financial measures of wealth and considering consumption levels in various income strata of society. Through this comprehensive analysis, we gain valuable insights into the different approaches and the real-life affordability experiences of smokers. It highlights the importance of considering multiple factors in understanding cigarette affordability and its implications for public health and tobacco control policies, thereby providing valuable insights for future policy decisions, not only in Malaysia but also other countries worldwide.

The study is subject to several limitations. Firstly, HIPC and HEPC were produced from survey data which may lead to predisposition, as the surveys do not cover residential institutions (e.g., hotels, hostels, hospitals, and senior citizens' home) which might slightly bias the data and hence the results herein. They were also not available on a yearly basis preventing analysis in all years. Furthermore, the surveys used to generate estimates of HIPC, HEPC, and consumption levels were presented as point estimates only. No confidence intervals were provided (even in the accompanying technical reports) and as such, wider statistical analysis could not be performed.

The use of HIPC in our calculation of cigarette affordability does not consider the portion of income (benefits) received from the government. Moreover, HIPC also does not consider the use of disposable income, thereby disregarding the impact of various deductions and taxes that households may be subject to. Consequently, this approach may result in a slight overestimation of households' actual purchasing power when assessing the affordability of cigarettes. Nevertheless, the impact of this should be broadly consistent over time and hence our study still provides valuable insights into the trends and changes in cigarette affordability, and hence helps in understanding the broader dynamics of tobacco consumption and the effectiveness of tobacco control policies in Malaysia.

In our assessment of various financial measures of wealth, such as GDP, household income, and household expenditure, we have adopted a per capita approach. This method involves using the mean household size, which includes young children who do not contribute to household income or engage in smoking behaviour. Consequently, their inclusion in these calculations may potentially present an inaccurate depiction of the financial dynamics within households. For example, consider a family consisting of two adults and two children. Per capita calculations would imply that the family has access to an amount equivalent to four times per capita GDP, which is unlikely to accurately reflect the actual financial resources available, hence may overestimate or underestimate affordability, depending on the circumstances. It is important to note, however, this issue is consistent across all the measures of wealth we have employed, and thus, does not affect the comparability between these indicators.

We do not fully allow for the fact that lower income people who smoke may adjust their individual consumption patterns when the overall population does not (or may do so

differently), as we only provided an overall estimate of consumption for the two population groups. Such behaviour might include switching to illicit sources not covered herein, or by reducing the number of sticks smoked per day. To allow for such possibilities we produced ultra-low estimates of consumption as part of our study (see Supporting Information).

Next, we acknowledge that using alternatives such as HIPC and HEPC instead of GDP per capita, and indeed consumption estimates, indicate that cigarettes require a greater share of the financial measures than the traditional RIP (and hence seem less affordable) and this may therefore lead to possible resistance to future tax increases. However, since our primary focus in this analysis has been on examining the trend in changes therein and hence shifts in afford-ability levels, rather than the absolute values or percentage points themselves, we suggest that the issue of resistance to tax increases becomes less pertinent to our study.

Finally, we specifically examined the affordability measures using the consumption levels of the smokers' population (while offering the calculations for the overall country population in the supplementary material). This distinction is essential as it allows us to discern the unique economic challenges faced by individuals or households where smoking is a prevalent habit. The calculation for the smokers' population provides targeted insights into the economic bur-den of cigarette consumption on those who engage in this behaviour. Furthermore, by using HIPC and HEPC, we tailor our analysis to the specific financial circumstances of households where smoking is a factor. This approach has allowed a more granular understanding of the financial strain imposed by cigarette consumption and has hence allowed nuanced policy rec-ommendations and interventions that address the unique challenges faced by the people who smoke.

## Conclusion

Understanding cigarette affordability is vital for assessing the efficacy of current government tobacco control measures, such as taxation, and hence how much measures can best be updated [2, 9, 39]. When assessing cigarette affordability, it is important to consider various financial measures of wealth, cigarette consumption estimates and socio-economic disparities among the population. By doing so, we can comprehensively understand the cigarette afford-ability patterns over time and the actual financial burden on the people who smoke. This is crucial as it also promotes better understanding in the potential impact of tobacco tax policies that consider the circumstances of the different groups [10, 16, 66].

Although the GDP per capita is a suitable starting point for getting a general overview of affordability, it does not provide sufficient details on household income growth, expenses, and income inequalities across different income groups as offered by the HIPC and HEPC mea-sures. Furthermore, exploring measures such as actual cigarette consumption provides further important insights, particularly in changes in smoking behaviour.

Our findings emphasise the need for action by the government to strengthen tobacco tax policies in Malaysia, identify potential gaps and opportunities for improvement, including multi-year tax strategies; stronger enforcement; border protection; and implementation of measures across the WHO Illicit Trade Protocol to effectively control the illicit cigarette trade and hence make tax policy more effective. Furthermore, the suggestions provided following our approach can be applied in other countries to improve the methodological insights into the cigarette affordability measures globally.

## Supporting information

**S1 Table. Yearly GDP per capita.**
(DOCX)

**S2 Table. Monthly Household Income Per Capita (HIPC) and Household Expenditure Per Capita (HEPC).**
(DOCX)

**S3 Table. Consumption estimates per person/year for 2011–2019 in rural areas.**
(DOCX)

**S4 Table. Consumption estimates per person/year for 2011–2019 in urban areas.**
(DOCX)

**S5 Table. Affordability calculation using conventional method (2000RIP).**
(DOCX)

**S6 Table. Affordability calculation using actual consumption estimates (Consumption-RIP) based on overall population.**
(DOCX)

**S7 Table. Affordability calculation using conventional method (2000RIP).**
(DOCX)

**S8 Table. Affordability calculation using actual consumption estimates (Consumption-RIP) based on smokers population.**
(DOCX)

**S1 Data. Raw data file.**
(XLSX)

## Acknowledgments

The authors would like to thank all the parties who helped identify the data sources. The responsibility for any errors remains entirely with the authors.

## Author Contributions

**Conceptualization:** Ridzuan Kunji Koya, J. Robert Branston, Allen W. A. Gallagher.

**Data curation:** Ridzuan Kunji Koya.

**Formal analysis:** Ridzuan Kunji Koya.

**Funding acquisition:** Ridzuan Kunji Koya.

**Project administration:** Ridzuan Kunji Koya.

**Supervision:** J. Robert Branston, Allen W. A. Gallagher.

**Writing – original draft:** Ridzuan Kunji Koya, J. Robert Branston, Allen W. A. Gallagher.

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
