## [Decision Letter · Decision Letter 0]

1 May 2024

PONE-D-24-02598Exploring the Value in Variations of the Relative Income Price (RIP) for Calculating Cigarette Affordability: An Illustration using MalaysiaPLOS ONE

Dear Dr. Branston,

Thank you for submitting your manuscript to PLOS ONE. After careful consideration, we feel that it has merit but does not fully meet PLOS ONE’s publication criteria as it currently stands. Therefore, we invite you to submit a revised version of the manuscript that addresses the points raised during the review process.

We look forward to receiving your revised manuscript.

Kind regards,

Vanessa Carels

Staff Editor

PLOS ONE

[JRB owns 10 shares in Imperial Brands for research purposes.  The shares were a gift from a public health campaigner and are not held for financial gain or benefit.  All dividends received are donated to health related charities, and proceeds from any future share sale or takeover will be similarly donated.]. 

2. We note that your Data Availability Statement is currently as follows: [All data used herein is publicly available and obtained from the sources listed within the manuscript.]

Reviewers' comments:

Reviewer's Responses to Questions

**Comments to the Author**

1. Is the manuscript technically sound, and do the data support the conclusions?

Reviewer #1: Partly

Reviewer #2: Partly

Reviewer #3: Partly

2. Has the statistical analysis been performed appropriately and rigorously? 

Reviewer #1: Yes

Reviewer #2: Yes

Reviewer #3: Yes

3. Have the authors made all data underlying the findings in their manuscript fully available?

Reviewer #1: Yes

Reviewer #2: Yes

Reviewer #3: Yes

4. Is the manuscript presented in an intelligible fashion and written in standard English?

Reviewer #1: Yes

Reviewer #2: Yes

Reviewer #3: Yes

5. Review Comments to the Author

Reviewer #1: The authors propose a method to measure cigarette affordability, to improve the classical relative income price (RIP) based on GDP per capita (Blecher & van Walbeek, 2004). The method is based on household income per capita (HIPC) and household expenditure per capita (HEPC). The new measure suggests that a stronger tax increase is needed for Malaysia than the calculations based on the GDP-RIP.

As mentioned in the document, the idea was used before for China (16) and Russia (8), in order to allow for a measure that captures heterogeneity across the income distribution. They base their analysis on the disposable household income per capita; several categories are made in or. Their measures kept the 20cigs x 100 packs, for specific sets of brands, as the standard baskets of consumption. The authors contributions are (i) to consider expenditures rather than income, (ii) to define a ‘moving’ basket (central consumption estimates).

In my opinion, the RIP was designed for comparability across countries as a simple measure that provides a rapid answer to the question: are our cigarettes expensive or not? By adding the heterogeneity extension, is the possible to state if the cigarettes are expensive or not for a given set of the population. The concept behind this paper is a bit more complex as it aims to be micro-founded on the decisions to quit or change consumption. In other words, it is linked to elasticities per se, then it is subject to all the endogeneity considerations of this statistical object: there is selection on who smoke, and what do they smoke, and how frequent. Those are the implications of deciding to allow for a basket that is dependent on the information of the survey; even if it is used as an aggregate. Such limitations should be acknowledged in the discussion section.

The authors might consider keeping the basket fixed and define an additional concept that is linked to the peer-reviewed consumption elasticities estimates for the country. In this way, it is easier to contribute to the discussion without the hazard of using constructs that are a direct function of the price.

An important issue is the lack of standard errors. The HIPC and HEPC estimates are based on survey data. The authors can perform a simple ‘box’ exercise for the 2000RIP estimates (i.e., use the lower and upper bound of the CI for income), but for the ConsumptionRIP a bootrapping strategy is required as both the numerator and the denominator are constructed with survey data. This is central to be able to say that there are actual differences between the strategies considered.

Minor comments

HEPC acronym should be included at the same time as the HIPC

References

8. Zheng R, Marquez PV, Kuznetsova P, Hu X, Wang Y. Cigarette Affordability in the Russian

Federation 2002-2017 [Available from: https://elibrary.worldbank.org/doi/abs/10.1596/30620.

16. Nargis N, Zheng R, Xu SS, Fong GT, Feng G, Jiang Y, et al. Cigarette Affordability in China, 2006–2015: Findings from International Tobacco Control China Surveys. International Journal of Environmental Research and Public Health. 2019;16(7).

Reviewer #2: I think the paper is interesting. The discussion of tobacco taxation and health impacts is quite relevant. However, I have some concerns about methodological choices to obtain the affordability measures.

1 - Why choose a fixed annual consumption of 2,000 sticks? What is the rationale on this? Is it the same for urban and rural areas? If you have a survey on that you should used and be clear on this. I am quite confused on how this went on.

2 - I do not think the measure you are using is indeed an affordability measure. If we think in household level it is ok. But, in a agreggated form I do not think so. It looks like a productivity difference between the tobacco industry and the average income worker (using gdp per capita, income or expenditure). If this is true, the affordability is expected to drop for most countries, since the tobacco industry is more productive than the average worker sector for most coutries. This is my main concern.

Reviewer #3: First, I want to thank the authors and the editor for the opportunity to review this paper. The study reports a detailed timeline of cigarette affordability using diverse methods to measure affordability. Recommending using household/income expenditure per capita instead of GDP measures allows for analysis of the change in affordability across different demographic groups which would be useful to improve tobacco control policies. However, the measures using yearly consumption of cigarettes are not comparable across years as endogeneity exists between price and consumption. I would recommend addressing this issue before being ready for publication.

My specific comments are shown below.

Comment 1: If consumption changes between years, then the estimates are not comparable across years. A decrease in ConsumptionRIP can be caused by (i) a decrease in consumption, (ii) a price decrease, and (iii) an increase in household income. The relationship between consumption and price leads to an endogeneity problem, as increases in price during the year i would decrease consumption (by the price elasticity), which the authors are aware of. With 2,000 cigarettes (or any other constant quantity), a price increase would reduce affordability:

2000RIP_HEPC_i=(2000·↑Price_i)/HEPC_i

However, in the ConsumptionRIP measures, the measure would change something like this:

ConspumptionRIP_HEPC_i=↓Cons·↑Price_i)/HEPC_i

If the ConsumptionRIP increases or decreases then will be determined by the price elasticity; if it’s inelastic, the consumption would decrease less than the price increase, so it would become less affordable, but if it's elastic then it would be more affordable.

Because of these interpretation issues, I don’t recommend using ConsumptionRIP for the analysis of affordability. What it would be useful to interpret is to define a year to use as a baseline. For example, set at the Consumption level of 2009. That way the interpretation would be if it’s less or more affordable to buy the average consumption of 2009.

Comment 2: Does the law require to adjust the specific tax by the inflation every X years? Does the increase happen automatically or a new law is required to adjust taxes by inflation? An effective way to ensure adjusting by inflation is to include it in the tobacco law.

Comment 3: In the discussion section, it is mentioned that the differences are significant. Was a test used to assess the significance? It should be stated which test and significance level. If no test was used, I recommend changing the wording.

Comment 4: The use of household expenditure and income is a great idea to measure how affordability changes across income groups. In the introduction was mentioned that one of the advantages of HEPC/HIPC measures is the analysis by income groups, but this was not done (apart from urban and rural was expecting that type of analysis and can be an important contribution that doesn’t require much more work than doing the same analysis as urban and rural, but, for example, by deciles of HEPC/HIPC. Has the changes in affordability been different for lower and higher-income groups?

6. PLOS authors have the option to publish the peer review history of their article (what does this mean?). If published, this will include your full peer review and any attached files.

Reviewer #1: No

Reviewer #2: No

Reviewer #3: **Yes: **Daniel Araya-Rios

---

## [Author Response · Author response to Decision Letter 0]

11 Jul 2024

We thank the editor and reviewers for their kind comments and for the very helpful suggestions they made for how we can improve the paper. It was gratifying that all reviewers found value in the paper and were generally very supportive of it. We have made some revisions to the paper in light of the helpful comments received, and provide further details of our response to each comment received in the paragraphs below in red.

Editor’s comment:

1) Please ensure that your manuscript meets PLOS ONE's style requirements, including those for file naming. The PLOS ONE style templates can be found at:

Thank you for the reminder to follow PLOS ONE’s style requirements. We have ensured that our manuscript meets these requirements, including proper file naming.

[JRB owns 10 shares in Imperial Brands for research purposes. The shares were a gift from a public health campaigner and are not held for financial gain or benefit. All dividends received are donated to health related charities, and proceeds from any future share sale or takeover will be similarly donated.]. 

Thank you for your instructions regarding the Competing Interests section. There are no restrictions on the sharing of data and/or materials. Therefore we have updated our Competing Interests statement accordingly and included it in our cover letter as below:

Regarding my competing interest statement we can confirm that “This does not alter our adherence to PLOS ONE policies on sharing data and materials” and that we are happy for you to update our online submission form in this regard.

3. We note that your Data Availability Statement is currently as follows: [All data used herein is publicly available and obtained from the sources listed within the manuscript.]

Thank you very much for your queries regarding data availability. We confirm that our submission contains all raw data required to replicate the results of our study, which are presented as the "minimal data set" in the Supporting Information as “File S9: Raw Data used for Study Results.” This file contains individual tabs representing the tables and figures used in the study.

Specifically, we have submitted the following data:

• The values behind the measures reported (means, standard deviations, etc.)

• The values used to build graphs

• The points extracted from images for analysis

Additionally, we have provided the supporting data as Supporting Information (Table S1 to Table S8) files.

We also confirm no ethical or legal restrictions on sharing a de-identified data set. All data used herein were produced by Malaysian government entities and are publicly available.

Reviewer 1’s comments:

The authors propose a method to measure cigarette affordability, to improve the classical relative income price (RIP) based on GDP per capita (Blecher & van Walbeek, 2004). The method is based on household income per capita (HIPC) and household expenditure per capita (HEPC). The new measure suggests that a stronger tax increase is needed for Malaysia than the calculations based on the GDP-RIP.

As mentioned in the document, the idea was used before for China (16) and Russia (8), in order to allow for a measure that captures heterogeneity across the income distribution. They base their analysis on the disposable household income per capita; several categories are made in or. Their measures kept the 20cigs x 100 packs, for specific sets of brands, as the standard baskets of consumption. The authors contributions are (i) to consider expenditures rather than income, (ii) to define a ‘moving’ basket (central consumption estimates).

We are glad to see that our writing was able to convey clearly the key points we wanted to explore in the paper. We are sincerely grateful for your thoughtful feedback and comments, which we address individually in the paragraphs below.

1) In my opinion, the RIP was designed for comparability across countries as a simple measure that provides a rapid answer to the question: are our cigarettes expensive or not? By adding the heterogeneity extension, is the possible to state if the cigarettes are expensive or not for a given set of the population. The concept behind this paper is a bit more complex as it aims to be micro-founded on the decisions to quit or change consumption. In other words, it is linked to elasticities per se, then it is subject to all the endogeneity considerations of this statistical object: there is selection on who smoke, and what do they smoke, and how frequent. Those are the implications of deciding to allow for a basket that is dependent on the information of the survey; even if it is used as an aggregate. Such limitations should be acknowledged in the discussion section.

We agree with you that the Relative Income Price (RIP) was designed to provide a straightforward metric as it compares the cost of 2,000 cigarette sticks as a proportion of GDP, allowing for temporal and international comparisons. This is something that is very valuable but also inherently limited, so, we have introduced additional complexities to this framework in order to develop further insights.

As you note one implication of the complexities we have introduced by considering actual consumption levels, is the issue of endogeneity. We did this because we felt it was helpful to consider the typical cost of being a person who smokes. As such we are looking to explore both the affordability of tobacco (the RIP using the fixed 2,000 sticks) but also the affordability of being a typical smoker at different points in time (the RIP with the actual consumption estimates). 

Endogeneity is a common concern, particularly in econometric regression analyses, because it can lead to biased estimations. In such cases, the coefficients derived may be inaccurate and not reflect true relationships within the data (see for example, https://www.sciencedirect.com/topics/psychology/endogeneity). Since our study does not involve regression analysis, this concern does not apply.

What our approach does mean is that the versions of the RIP featuring actual consumption estimates are in practice exploring the overall impact of consumer behaviour, prices, and incomes/wealth, whereas the traditional RIP only explores prices and incomes. The endogeneity concern is therefore that consumer behaviour is in part determined by prices. With higher prices, the well established inelastic nature of tobacco products means that consumers will respond to some extent with slightly lower levels of consumption. As such what our traditional 2,000 stick RIP and our consumption RIP are allowing is an exploration of the impact of this consumer behaviour, as that is the cause of the difference between the different versions of the RIP. We think this is valuable in a market like Malaysia where there are different consumption levels in different parts of the economy, and where levels of illicit are high. This mean that many consumers might not be exposed to the price of legal cigarettes, or might be exposed to a different extent, and such detail wouldn’t be possible to explore with a single elasticity value (please also see our response below to your next point about the use of elasticity estimates).

Our study is therefore looking to offer a range of insights into the issue of affordability with the range of RIP series we provide, rather than rely solely on a single metric. Omitting the complexities associated with the versions of the RIP would simplify the analysis but at the cost of losing important details that contribute to a thorough understanding of tobacco affordability dynamics.

To outline this rationale, we have adjusted the methodology section of the revised paper (see p.10) where we now incorporate a new section entitled “Endogeneity between price and consumption levels”. This section outlines the potential endogeneity between price and consumption levels when examining the relationship between cigarette consumption and price. The section also explains the use of different RIP series in response to price changes in Malaysia, considering the substantial presence of an illicit tobacco market. 

We understand your suggestion to include this point in the limitations section. However, we chose to incorporate it into the methods section so that we can capture the complexity of the issue early on. By detailing it in the methods, we aimed to provide a more comprehensive understanding of the approach and its underlying intricacies, ensuring that the nuances are appropriately conveyed to the readers. We also note that the discussion section already reflects back on the endogeneity issue (p.22) so it should be something the reader is now fully able to consider holistically from the main body of the paper.

2) The authors might consider keeping the basket fixed and define an additional concept that is linked to the peer-reviewed consumption elasticities estimates for the country. In this way, it is easier to contribute to the discussion without the hazard of using constructs that are a direct function of the price.

Thank you for your suggestion. However, the country's most recent and only elasticity estimates were produced for the year 2009 and were published in 2013 (See ‘An Optimal Cigarette Tax in Malaysia’ by MN, Norashidah et al. http://www.ijem.upm.edu.my/vol7no2/bab02.pdf). As such this data is outdated, as the market has seen a consider increase in the illicit market during this time, with our own work showing it has gone from 40.7% in 2011 to 69.9% in 2019 – see https://doi:10.1136/tc-2023-058333.

As noted above in our response to your first comment, we believe there is value in presenting a range of different version of the RIP including those with both fixed consumption volumes, and those based on actual consumption estimates. In order to better explain the value of this approach we have made the various changes as outlined in that answer, most notably in the new section on endogeneity in the methodology section.

We hope this will help readers understand why we present such a variety of data series and see the value in them. However, we note that including versions that allow consumption to changes doesn’t in any way impact the versions of the RIP with the fixed 2000 stick consumption level. As such any reader who remains concerned by the endogeneity issue will be able to concentrate on the traditional RIP measures if they so wish.

3) An important issue is the lack of standard errors. The HIPC and HEPC estimates are based on survey data. The authors can perform a simple ‘box’ exercise for the 2000RIP estimates (i.e., use the lower and upper bound of the CI for income), but for the ConsumptionRIP a bootrapping strategy is required as both the numerator and the denominator are constructed with survey data. This is central to be able to say that there are actual differences between the strategies considered.

Thank you for your valuable feedback. Unfortunately, the government reports from which the HIPC, HEPC, and consumption data was sourced presented only point estimates. The report from which HIPC and HEPC was sourced does have an accompanying technical note that acknowledges the potential for sampling and non-sampling errors. However, it did not provide confidence intervals (except for one illustrative example) to account for these errors. The technical note accompanying the consumption data adopts the same approach. Therefore, given the lack of confidence intervals and associated information, we were unable to conduct a wider statistical analysis in the manner you have suggested. 

To account for this inherent weakness in the survey derived data utilised, we have expanded the limitations section to cover the point. The paragraph covering survey limitations now (p.26) reads as:

“The study is subject to several limitations. Firstly, HIPC and HEPC were produced from survey data which may lead to predisposition, as the surveys do not cover residential institutions (e.g., hotels, hostels, hospitals, and senior citizens’ home) which might slightly bias the data and hence the results herein. They were also not available on a yearly basis preventing analysis in all years. Furthermore, the surveys used to generate estimates of HIPC, HEPC, and consumption levels were presented as point estimates only. No confidence intervals were provided (even in the accompanying technical reports) and as such, wider statistical analysis could not be performed.”

4) Minor comments

HEPC acronym should be included at the same time as the HIPC

We appreciate your thorough review and attention to detail. We apologise that we did not include the HEPC acronym in the abstract section where it was first introduced, and we have now corrected this error. We have also carefully checked the rest of the manuscript with aim of ensuring that all acronyms are consistently identified on first use throughout the manuscript.

Reviewer 2’s comments:

I think the paper is interesting. The discussion of tobacco taxation and health impacts is quite relevant. However, I have some concerns about methodological choices to obtain the affordability measures.

We are glad that you find this paper interesting. In the paragraphs below we address the specific queries raised.

1) Why choose a fixed annual consumption of 2,000 sticks? What is the rationale on this? Is it the same for urban and rural areas? If you have a survey on that you should used and be clear on this. I am 

quite confused on how this went on.

Thank you for your comment regarding the selection of the fixed annual consumption of 2,000 sticks. This choice is simply our adopting the conventional RI

---

## [Decision Letter · Decision Letter 1]

12 Aug 2024

PONE-D-24-02598R1Exploring the Value in Variations of the Relative Income Price (RIP) for Calculating Cigarette Affordability: An Illustration using MalaysiaPLOS ONE

Dear Dr. Branston,

Thank you for submitting your manuscript to PLOS ONE. After careful consideration, we feel that it has merit but does not fully meet PLOS ONE’s publication criteria as it currently stands. Therefore, we invite you to submit a revised version of the manuscript that addresses the points raised during the review process.

We look forward to receiving your revised manuscript.

Kind regards,

Nik Ahmad Sufian Burhan

Academic Editor

PLOS ONE

Journal Requirements:

Reviewers' comments:

Reviewer's Responses to Questions

**Comments to the Author**

1. If the authors have adequately addressed your comments raised in a previous round of review and you feel that this manuscript is now acceptable for publication, you may indicate that here to bypass the “Comments to the Author” section, enter your conflict of interest statement in the “Confidential to Editor” section, and submit your "Accept" recommendation.

Reviewer #2: All comments have been addressed

Reviewer #3: All comments have been addressed

2. Is the manuscript technically sound, and do the data support the conclusions?

Reviewer #2: Yes

Reviewer #3: Yes

3. Has the statistical analysis been performed appropriately and rigorously? 

Reviewer #2: Yes

Reviewer #3: I Don't Know

4. Have the authors made all data underlying the findings in their manuscript fully available?

Reviewer #2: Yes

Reviewer #3: Yes

5. Is the manuscript presented in an intelligible fashion and written in standard English?

Reviewer #2: Yes

Reviewer #3: Yes

6. Review Comments to the Author

Reviewer #2: Thanks for the opportunity to review the paper again. I am satisfied with the authors answers and modification. In this way the new manuscript should be accepted for publication in the journal.

Reviewer #3: Dear authors,

Thanks for your response to my comments and for addressing all of them.

I have no major comments regarding my previous comments. I still think that consumption measures can be misinterpreted without a closer look at the change in average consumption and household expenditure and income. However, the authors provide this information (in the paper tables and supplemental material. Moreover, in the discussion section, the authors interpret the consumption measures in light of the changes in consumption and income.

I recommended a minor revision only because of one issue. If I understand correctly, the consumption HIPC and HEPC measures can be interpreted as the (annual) budget share of households dedicated to tobacco. If I am correct, these budget shares seem too large, suggesting that, in 2019, 25.9% of all household expenses (HEPC) were dedicated to tobacco products (and 40.5% in rural households). It is possible that these numbers are correct and that (smokers) rural households devote 40.5% of their expenses to tobacco, but these would be a lot higher than what the literature has found. A paper titled "Distribution of Household Tobacco Expenditure and Household Affordability of Tobacco Products in Malaysia" measured the budget share of tobacco products in Malaysia using the Household Expenditure Surveys and found that the overall budget share for tobacco products was 4.6%. Looking at other countries' estimates, the budget shares are closer to 5% than 26%.

These differences should be addressed in the paper, as indicating that the average rural smoker devotes 40.5% of their expenses to tobacco products is a "big" statement.

7. PLOS authors have the option to publish the peer review history of their article (what does this mean?). If published, this will include your full peer review and any attached files.

Reviewer #2: No

Reviewer #3: **Yes: **Daniel Araya-Ríos

---

## [Author Response · Author response to Decision Letter 1]

28 Oct 2024

Journal Requirements:

We have carefully reviewed the reference list to ensure its accuracy and completeness. We confirm that no retracted papers have been cited or used in the manuscript. As such, no changes to the reference list are necessary.

Reviewer #3 comment:

Thanks for your response to my comments and for addressing all of them.

I have no major comments regarding my previous comments. I still think that consumption measures can be misinterpreted without a closer look at the change in average consumption and household expenditure and income. However, the authors provide this information (in the paper tables and supplemental material. Moreover, in the discussion section, the authors interpret the consumption measures in light of the changes in consumption and income.

We are glad the reviewer feels the information we have included in the revised paper and its supplementary materials is sufficient to address their previous comments. Again, we thank the reviewer for their constructive input.

I recommended a minor revision only because of one issue. If I understand correctly, the consumption HIPC and HEPC measures can be interpreted as the (annual) budget share of households dedicated to tobacco. If I am correct, these budget shares seem too large, suggesting that, in 2019, 25.9% of all household expenses (HEPC) were dedicated to tobacco products (and 40.5% in rural households). It is possible that these numbers are correct and that (smokers) rural households devote 40.5% of their expenses to tobacco, but these would be a lot higher than what the literature has found. A paper titled "Distribution of Household Tobacco Expenditure and Household Affordability of Tobacco Products in Malaysia" measured the budget share of tobacco products in Malaysia using the Household Expenditure Surveys and found that the overall budget share for tobacco products was 4.6%. Looking at other countries' estimates, the budget shares are closer to 5% than 26%.

These differences should be addressed in the paper, as indicating that the average rural smoker devotes 40.5% of their expenses to tobacco products is a "big" statement.

Thank you for your thoughtful feedback and for raising the concern regarding the size of the Household Expenditure per Capita (HEPC) and Household Income per Capita (HIPC) measures.

In order to be sure of our results we have carefully reviewed our data and calculations, and while we acknowledge that our income/expenditure estimates seem a bit high in comparison to existing literature, we can confirm that the values are correct given our dataset and methodological framework adopted. We therefore think these figures reflect the reality of rural smoker households, which may face substantial socio-economic pressures as a result. This could result in a higher income allocation to tobacco products than other needs which could be an important finding for politicians.

We appreciate your reference to the paper, "Distribution of Household Tobacco Expenditure and Household Affordability of Tobacco Products in Malaysia", which reports a national average tobacco budget share of 4.6%. Upon reviewing this paper, we note that the suggested figure isn’t directly comparable to the estimates in our paper. The 4.6% expenditure level is for the entire population of households whereas our study looks at the cost for individuals who use tobacco. Furthermore, in the aforementioned paper it says that the proportion of households reporting tobacco expenditure has never dropped below 35% which means that most households covered do not in fact spend any money on tobacco. As such a rough estimate would be that the expenditure in only those households who report tobacco consumption would be approximately three times the reported level or around 15%. While this is still lower than our own estimate of (for example) 26% for overall household expenses in 2019, it is far closer than the numbers might initially suggest. Furthermore, we report in our paper (p.16) that in the appendix we include more directly comparable estimates. For example, our expenditure based affordability estimates using actual consumption for the overall population in 2014 is 4.6% which is very much line with the other study. (NB in the paper on p.16 we report the version of this affordability that covers just the rural area, which is 6.4%). 

Our study specifically targets rural and urban smoker households, both of which will allocate a much higher proportion of their budgets to tobacco products compared to the general population. Socio-economic factors, such as lower incomes and higher smoking prevalence in rural areas, contribute to the higher expenditure on tobacco products for such groups. Furthermore, it is not unusual for studies focusing on particular subgroups, such as heavy smokers or rural households, to report higher tobacco-related expenditures for such groups relative to national averages. See for example: Rossouw, L. (2022) looking at the inequalities of socioeconomic status and tobacco consumption in China, Ghana, India, Mexico, the Russian Federation and South Africa; and Pahari, S. et al. (2023) looking at the meta-analysis of evidence drawn from regional studies of tobacco usage in India between 2010 and 2022.

As you rightly observed, the discussion section of our paper interprets these consumption measures in the context of changes in household income and consumption. The high HEPC values reported are an important finding, as they emphasise the economic burden of tobacco consumption on vulnerable subpopulations, particularly in rural areas. Although the 40.5% figure is a startling level to report, we believe it accurately captures the financial strain experienced by these households.

In order to recognise the above we have revised the paper in the discussion section (p.15) so that it now reads as: 

“When we employed 2000RIP with HIPC or HEPC as alternatives to GDP for measuring affordability changes in both rural and urban areas, the outcomes were noticeably different. Indeed, these alternate measures suggest that the amount of financial wealth required is notably larger than is suggested by standard versions of the RIP using GDP. Furthermore, the shift in affordability became more pronounced when we integrated the ConsumptionRIP with HIPC and HEPC.” 

Furthermore, we have also adjusted the discussion later on (P.16) where we refer the reader to the appendix with population level estimates, in order to highlight why the results in the main paper look considerably different to previously published studies in the first instance. In doing so we explicitly cite the paper from Tan et al. highlighted by the referee. This section now reads as:

“When looking the affordability as calculated using mean tobacco use for the entire country population (see S6), we find a relatively lower level of the RIP. This suggests that cigarettes are more affordable when considering the overall population, in contrast to calculations based solely on the smokers’ population. For instance, our ConsumptionRIP-HEPC-Rural, when applied to the overall population, revealed that the percentage of expenditure required to purchase cigarettes was 6.4% in 2014, compared to 43.8% within the smokers’ population. By 2016, these figures had increased to 6.9% and 47.3% respectively. Such differences explain why the results herein might look considerably different from previously published studies on first inspection (see for example, the results in a paper by Tan et al. (2020), which measured the budget share of tobacco products in Malaysia using the Household Expenditure Surveys (HES) and found that the overall budget share for tobacco products was 4.6%)(37). Nevertheless, when considering the changes in the two affordability measures between dates, the aforementioned illustration shows there is broad consistently between the measures. Between 2014 and 2016 there is a change in affordability of 7.8% (6.4% to 6.9%) using the overall population but 7.99% within the smokers’ population (43.8% to 47.3%).”

---

## [Editor Report · Decision Letter 2]

30 Oct 2024

Exploring the Value in Variations of the Relative Income Price (RIP) for Calculating Cigarette Affordability: An Illustration using Malaysia

PONE-D-24-02598R2

Dear Dr. Branston,

We’re pleased to inform you that your manuscript has been judged scientifically suitable for publication and will be formally accepted for publication once it meets all outstanding technical requirements.

Kind regards,

Nik Ahmad Sufian Burhan

Academic Editor

PLOS ONE
---

## [Editor Report · Acceptance letter]

6 Nov 2024

PONE-D-24-02598R2 

PLOS ONE

Dear Dr. Branston, 

I'm pleased to inform you that your manuscript has been deemed suitable for publication in PLOS ONE. Congratulations! Your manuscript is now being handed over to our production team.

Kind regards, 

on behalf of

Dr. Nik Ahmad Sufian Burhan 

Academic Editor

PLOS ONE